# Co_3_O_4_ Nanoneedle Array Grown on Carbon Fiber Paper for Air Cathodes towards Flexible and Rechargeable Zn–Air Batteries

**DOI:** 10.3390/nano11123321

**Published:** 2021-12-07

**Authors:** Ziyuan Li, Wenjia Han, Peng Jia, Xia Li, Yifei Jiang, Qijun Ding

**Affiliations:** State Key Laboratory of Biobased Material and Green Papermaking, Qilu University of Technology, Shandong Academy of Sciences, Jinan 250353, China; ziyuanli97@126.com (Z.L.); hwj200506@163.com (W.H.); milai19871219@163.com (P.J.); sqlixia126@126.com (X.L.)

**Keywords:** flexible zinc-air battery, papermaking, cobalt oxides, carbon fiber, hydrothermal method

## Abstract

An economical and efficient method is developed for preparing flexible cathodes. In this work, a dense mesoporous Co_3_O_4_ layer was first hydrothermally grown in situ on the surface of chopped carbon fibers (CFs), and then carbon fiber paper (Co_3_O_4_/CP) was prepared by a wet papermaking process as a flexible zinc-air battery (ZAB). The high-performance air cathode utilizes the high specific surface area of a single chopped carbon fiber, which is conducive to the deposition and adhesion of the Co_3_O_4_ layer. Through the wet papermaking process, Co_3_O_4_/CP has ultra-thin, high mechanical stability and excellent electrical conductivity. In addition, the assembled ZAB exhibits relatively excellent electrochemical performance, with a continuous cycle of more than 180 times at a current density of 2 mA·cm^−2^. The zinc-air battery can maintain a close fit and work stably and efficiently even under high bending conditions. This process of combining single carbon fibers to prepare ultra-thin, high-density, high-conductivity carbon fiber paper through a papermaking process has huge application potential in the field of flexible wearables.

## 1. Introduction

With the improvement of living standards and the development of society, people have higher requirements for environmental protection and convenience of energy utilization [1,2,3]. At present, secondary energy storage batteries, as a kind of sustainable and recycled clean energy, have been able to replace most of the functions of fossil fuels and are widely used in smart devices, automobiles, aerospace and other fields [4,5]. In recent years, the optimization of secondary energy storage batteries in terms of specific capacity, cycle life, rate performance, etc. has attracted the attention of researchers [6,7]. The emergence of flexible electronic devices such as foldable mobile phones and flexible electronic screens also puts forward high energy density and flexibility requirements for batteries [8,9,10]. Zinc-air battery (ZAB), as one of the secondary batteries, has excellent performance such as high specific energy and current density, making it an ideal candidate for flexible energy storage equipment [11,12,13]. However, the current mainstream anode material of ZAB is commercial graphite, which has poor cycle performance and is far below the standard for flexible devices [14,15]. Therefore, there is an urgent need to replace graphite with excellent flexible negative electrode materials.

Carbon fiber (CF), as a flexible inorganic fiber material with a high carbon content, has become the best material to replace commercial graphite electrodes due to its high tensile strength, good electrical conductivity and excellent flexibility [16,17,18]. Xu et al. assembled a fiber-type zinc-carbon battery using a single carbon fiber as the electrode through a flexible plastic tube to seal the electrolyte [19]. However, the reduction of carbon content and the addition of organic matter inevitably led to the decline of the electrochemical performance of carbon fibers [20]. At present, traditional precious metal catalysts such as Au and Ru are gradually being replaced by non-precious metal catalysts due to the lower cost and abundant reserves [21]. Charles et al. compared and analyzed the catalytic effects of each element by depositing Ni, Co, Ir oxide and metal compound catalytic layers on the graphene surface [22]. The results show that under the condition of alkaline electrolyte, every non-precious metal system has a certain catalytic effect. Therefore, the performance and chargeability of flexible ZAB largely depends on the dual-function catalyst at the air cathode [23,24,25]. In addition, as a resource-rich gas, CO_2_ has a certain oxidation ability. It can etch micropores on the surface of carbon fibers at high temperatures, increase the specific surface area of carbon fibers, and is an excellent process to improve the surface activity of carbon fibers [26,27,28].

Among the dual-functional catalysts, nanostructured Co_3_O_4_ has become one of the most promising catalytic materials due to its nanoporous structure, abundant sources, excellent catalytic activity and alkali resistance [29,30]. Liang et al. grew and synthesized Co_3_O_4_ nanocrystals on reduced graphene oxide, which proved that carbon-based materials and Co_3_O_4_ have high oxygen reduction reaction (ORR) activity after hybridization [31]. Bai et al. studied the Co_3_O_4_ supported structure and proved that the Co_3_O_4_ layer under the 3D structure has a higher specific surface area and good mesoporous characteristics, and better oxidation ability [32]. However, there are few reports on the application of Co_3_O_4_ loading in ZAB. Most research focuses on loading directly onto traditional two-dimensional planar structures such as carbon cloth and carbon felt with huge or planar structures. For example, Chen et al. deposited Co(OH)_2_ on the surface of carbon cloth by electrophoretic deposition, and prepared a Co_3_O_4_ catalytic layer by oxidation annealing, which effectively improved the catalytic performance of the air cathode [33]. Zhang et al. prepared a new type of multifunctional catalyst by carbonizing a composite material composed of an imidazole molecular sieve framework and carbon fiber paper, which showed good catalytic activity [34]. It is worth noting that although the two-dimensional structure load achieved a certain catalytic effect, the overall load difficulty is higher than that of the one-dimensional structure. This is due to the decrease in the specific surface area of the carbon fiber due to the oriented multilayer arrangement structure [35,36]. At the same time, high-temperature treatments such as carbonization also require the two-dimensional material itself to have properties such as heat resistance and high bonding strength, and have a high cost. Li et al. developed a bifunctional catalyst composed of atomically layered mesoporous cobalt/nitrogen-doped reduced graphene oxide nanosheets, and prepared a fibrous zinc-air battery, which showed a stable electrochemical performance [37]. Unfortunately, fibrous batteries have the problems of a small size and a low electric capacity [38,39,40]. Therefore, it is very critical to develop a simple and continuous one-dimensional carbon fiber catalytic layer loading and a two-dimensional integration process.

In this work, we used chopped carbon fibers as templates and carbon resources to successfully prepare Co_3_O_4_/CP air cathodes through hydrothermal reaction synthesis and wet molding processes. ZAB was assembled and prepared by a fast and simple lamination method. The increase in the specific surface area and active sites of the single chopped carbon fiber after the CO_2_ etching treatment leads to a greatly increased Co adhesion rate, which is beneficial to improve the electrocatalytic performance. When Co_3_O_4_/CP was used as the air cathode material, the prepared ZAB exhibited a high round-trip efficiency and a good charge–discharge cycle stability, and it was able to work stably in a variety of folded states. These ultra-thin paper-like composite electrode materials will help to further promote the development of ZAB in the field of various wearable and flexible energy storage, and have great potential.

## 2. Experimental Section

### 2.1. Preparation of Air Cathodes for ZAB

All chemicals with analytical grade were purchased from Sigma-Aldrich and were used directly. The air cathode was prepared as the following processes (Figure 1). Firstly, the carbon fibers (CFs) with the length of 3 mm (Toho Tenax, Tokyo, Japan) were modified via the CO_2_ etching at 500 °C for one hour, aiming at removing the impurities, improving the hydrophilicity and increasing specific surface area (Appendix A). Secondly, Co(OH)_2_ nanoneedles were grown on the surface of modified fibers (m-CFs) at 150 °C for 3 h in a Teflon-lined stainless-steel autoclave containing a hydrothermal solution of 120 mL with 0.02 M Co(NO_3_)_2_·6H_2_O and 0.01 M urea. Thirdly, the Co (OH)_2_/m-CFs were calcined at 300 °C for 2 h in a muffle furnace to obtain Co_3_O_4_/m-CFs. Finally, the Co_3_O_4_ supported on carbon fiber paper (Co_3_O_4_/CP) was produced from CFs by the wet papermaking process. Specifically, the water-based polyurethane was successively used as the dispersant and adhesive agent, and the wet Co_3_O_4_/CP was pressed at 95 psi and dried at 100 °C. The CP also was prepared according to the above processes for comparison.

### 2.2. Preparation of Hydrogel Electrolyte

The composite hydrogel electrolyte was prepared by a one-pot sol-gel method. Typically, 3.6 g of polyvinyl alcohol (PVA) powder was added to 30 mL of deionized water and stirred at 90 °C for 12 h to obtain a transparent and viscous PVA solution. Then, the glutaraldehyde (GA, 3.0 wt.% of PVA) and glycerol (GI, 2.0 wt.% of PVA) were added dropwise into the above PVA solution. Subsequently, dilute hydrochloric acid (1.0 wt.%) was added dropwise into the above solution till pH = 2 to promote the cross-linking reaction. Afterwards, the resulting solution was poured into a rectangular mold with a depth of 1 mm and was allowed to stand for 12 h for the sufficient cross-linking to become a hydrogel membrane. Finally, the hydrogel membrane was immersed into 6.0 M KOH for 24 h to obtain a hydrogel electrolyte membrane.

### 2.3. Fabrication of Flexible Zn-Air Battery

The flexible ZAB was assembled as a 4-layer structure from the below to above: polished and clean copper mesh, air cathode, hydrogel electrolyte membrane and polished and clean zinc foil (10 mm × 30 mm × 0.3 mm). Importantly, the breathable bandage was used for encapsulation and fixation.

### 2.4. Characterization

The surface morphologies and elemental mapping analyses of the air cathodes were characterized by a field-emission scanning electron microscope (SEM, Regulus 8220, Hitachi, Japan) equipped with an energy dispersive X-ray spectrometer (EDS, Hitachi, Japan). A high-resolution X-ray micro-computed tomography (Micro-CT, SkyScan 2211, Bruker, Berlin, Germany) was used to analyze the three-dimensional structures of the air cathode. An X-ray photoelectron spectrometer (XPS, ESCALAB Xi+, Thermo Fisher Scientific, Waltham, MA, USA) was used to analyze the elemental compositions and valences on the surface of air cathodes. The Raman analyses were performed on a laser confocal microscopy Raman spectrometer (Renishaw inVia, Gloucestershire, UK) with 532 nm laser excitation. The phase compositions were ascertained by using an X-ray diffractometer instrument (XRD, D8-ADVANCE, Bruker, Germany). The thermogravimetric (TG) analysis was carried out on a simultaneous thermal analyzer (STA449, Selb, Germany). The mechanical properties of commercial carbon paper (CP), CP and Co_3_O_4_/CP were tested at room temperature on a texture analyzer (TA. XT plusC, SMS, Surrey, UK). The prepared Co_3_O_4_/CP was subjected to multiple tensile tests and compared with commercial CP. The Young’s modulus is obtained by calculating the slope of the fitted straight line (fitted 5 times respectively, and the results are averaged) [41]. The air cathode was tested by using the four-point probe method.

An electrochemical workstation (CHI660E, Shanghai Chenghua Instrument Co., LTD., Shanghai, China) with a PINE rotating disk electrode (RDE) system was used to test the ORR activity of the Co_3_O_4_/CP material. Firstly, each air electrode material was ground into powder separately. Then, 4.0 mg of carbon powder material was ultrasonically dispersed in 2 mL of Nafion (5%), absolute ethanol, and deionized water (*v*:*v*:*v* = 1:1:4) solution. Then, 20 uL uniform ink was taken and loaded on to the glassy carbon (GC) electrode. Using platinum as the counter electrode and silver chloride as the reference electrode, the cyclic voltammetry scan test (CV) and the linear voltammetry scan test (LSV) were performed at a scan rate of 10 mV·s^−1^, and the voltage range is −1~0.2 V. The RDE test is measured in a 0.1 M KOH solution at room temperature under O_2_ saturation at different speeds ranging from 400 to 2025 rpm, and the scan rate is 10 mV·s^−1^. The battery charging and discharging polarization test was carried out by the line scan method, and the voltage range is 0~3 V. The number of electrons transferred per oxygen molecule at different potentials during the oxygen reduction reaction (ORR) (n) is calculated according to the Koutecky–Levich (K-L) equation [42]. The OER polarization curve test was carried out in 1.0 M KOH at room temperature, and the scan rate was 10 mV·s^−1^. The data representing OER activity were corrected for iR compensation.

The electrochemical properties of ZAB batteries were characterized on the Land battery test system (LAND CT2001A, Wuhan LAND Electronic Co., Ltd., Wuhan, China) by the polarization and galvanostatic charge/discharge test (GCD). Check the ZAB rate stability by changing the discharge current density to 0.05, 0.1, 0.5, 1, 2, 5, 1 mA·cm^−2^, and the discharge time is 10 min. The GCD test is carried out at a current density of 2 mA·cm^−2^, the charging and discharging time is 10 min each, and the protection voltage is set to 0.5~2.5 V.

## 3. Results and Discussion

As shown in Appendix A, after CO_2_ etching treatment, the hydrophilicity of the carbon fiber is significantly improved. Therefore, the EDS results show that the fiber surface is uniformly loaded with Co, O, and C elements (Appendix A). After annealing, Co_3_O_4_/CF maintained the high adhesion of the catalytic layer (Figure 1a). The EDS element mapping image also showed that Co (yellow), O (red) and C (blue) coexist and were evenly distributed on the carbon fiber’s surface. The growth area and density of the Co element were relatively high, and the nanosheet layered structure can be clearly observed on the surface of the carbon fiber, and had a high signal intensity of Co (Appendix A). After wet papermaking processes such as dispersion and suction filtration, an air cathode Co_3_O_4_/CP was obtained. The loading of the Co element remains high and uniform, and the Co_3_O_4_ catalytic layer showed a stable adhesion performance (Figure 1b).

Figure 2 shows the Micro-CT image of Co_3_O_4_/CP. The Co_3_O_4_/CF after hydrothermal growth and annealing presented a disorderly interlaced stacking state (Figure 2a,c). This multi-layer staggered structure enables a single carbon fiber to contact and bond efficiently in a three-dimensional space, which is beneficial to improve electrical conductivity. A small amount of resin adhesive was uniformly distributed in the form of small particles at the fiber interweaving points, ensuring the mechanical strength of Co_3_O_4_/CP without flocculation (Figure 2b). The highly loose pore structure of Co_3_O_4_/CP improved the electrochemical performance of flexible zinc-air battery such as rate, constant current charge and discharge.

In order to further prove the loading of the Co_3_O_4_ layer on the surface of the carbon fiber, X-ray photoelectron spectroscopy (XPS) tests were performed on the CF, Co(OH)_2_/CF, and Co_3_O_4_/CF in the process to observe their chemical composition and valence, as shown in Figure 3. The XPS spectra in Figure 3a,c show that the prepared Co(OH)_2_/CF and Co_3_O_4_/CF both contain Co, O and C elements, which are consistent with the EDS results. The C element comes from chopped carbon fiber, the XPS image of the original m-CF in Appendix A also proves this. Figure 3b shows the XPS spectrum curve of Co 2p in Co(OH)_2_/CF. The curve showed two main peaks at 796.5 and 780.2 eV, which correspond to the Co 2p_1/2_ and Co 2p_3/2_ orbitals of the metal ion Co. By peak fitting, the Co 2p_3/2_ peak in Figure 3d could be divided into two peaks, at 795.3 and 780.2 eV, indicating that the Co^2+^ and Co^3+^ states coexist in the Co_3_O_4_/CF sample. In addition, the binding energy difference (spin-orbit splitting) between the Co 2p_3/2_ and Co 2p_1/2_ peaks was 15.2 eV, which is also very consistent with the binding energy difference of pure Co_3_O_4_ reported in the literature [33].

As shown in Figure 4a, CF, Co(OH)_2_/CF and Co_3_O_4_/CF all exhibited two characteristic Raman peaks of carbon at 1350 cm^−1^ and 1596 cm^−1^. It corresponds to disordered carbon and sp^2^ carbon atom E_2g_ mode scattering, respectively. In addition, for Co_3_O_4_/CP and Co_3_O_4_/CF, the Raman spectrum clearly shows obvious peaks at 668 cm^−1^, and new peaks are also generated at 465 and 510 cm^−1^. These peaks are characteristic of Co_3_O_4_ [16,33,37]. The above results further proved that Co_3_O_4_ was successfully loaded on the surface of carbon fiber. By comparing the peak intensity of Co_3_O_4_/CF and Co_3_O_4_/CP, it is proved that Co_3_O_4_/CP still maintains a high catalytic layer loading. XRD measurements were performed on each sample, and the CF can only observe an obvious diffraction peak at the 2θ value of 25.4 (Figure 4b). The difference is that Co_3_O_4_/CP and Co_3_O_4_/CF have new diffraction peaks observed at 38.2, 45.4, 59.8, and 66.2, which proves the formation of the Co_3_O_4_ layer. This is also consistent with the above description. Since the formed Co_3_O_4_ supporting layer is thinner than the main body of carbon fiber, the XRD curve is still dominated by carbon fiber.

Due to the use of wet forming, hot pressing and drying, the structure of CP and Co_3_O_4_/CP is closer, the internal resistance is reduced (Appendix A), and the conductivity is improved. Appendix A shows and compares the cyclic voltammetry (CV) characteristic curve and the linear sweep voltammetry (LSV) curve of the air cathode before and after the Co_3_O_4_ load. From the CV curve, the electrode loaded with Co_3_O_4_ has significantly improved redox onset potential and peak current density than the unloaded electrode. The results show that the electrochemical performance of the oxidation-reduction reaction of Co_3_O_4_ supported by CP has been improved. Appendix A shows the OER curve measured by using commercial CP and Co_3_O_4_/CP electrode in 1.0 M KOH solution. Under the same current density, Co_3_O_4_/CP is significantly higher than commercial CP, showing certain OER activity. In addition, Co_3_O_4_/CP has a relatively low OER Tafel slope, indicating a favorable kinetic OER process. In order to show the ORR kinetics of the prepared air cathode, the ORR LSV of the Co_3_O_4_/CP electrode was measured at a different scan rate (Appendix A). According to the KL formula, the number of transferred electrons from −0.40 to −0.60 V to RHE is about 3.3 (Appendix A insert), which is relatively close to the current number of commercial catalytic transfer electrons, showing a certain catalytic performance [43]. This indicates the four-electron ORR mechanism of the Co_3_O_4_/CP sample.

As shown in Figure 5a, the Co_3_O_4_/CP electrode exhibits excellent bendability and flexibility. Figure 5b displays TG curves of the CF, Co_3_O_4_/CP and the commercial CP. All the curves show two evident weight losses. The first occurred at 200 °C, which means that the binder starts to decompose. The rapid weight loss at around 420 °C is mainly due to the complete pyrolysis of the binder waterborne polyurethane adhesive and the pyrolysis inside the carbon fiber. According to the TG curve comparison between CP and Co_3_O_4_/CP samples, the loading of Co_3_O_4_ accounts for 1.5% of the total carbon fiber mass. In addition, by comparing with commercial CP, carbon fiber paper prepared by papermaking method maintains a higher heat resistance while using less adhesive. The tensile stress–strain curves of Co_3_O_4_/CP and commercial CP are shown in Figure 5c. The Co_3_O_4_/CP have a significant improvement effect in stress intensity and tensile deformation properties compared to commercial CP. Co_3_O_4_/CP can reach a maximum stress of 42.5 MPa and a strain of 2.25%. By comparing Co_3_O_4_/CP with CP, it was found that the loading of the catalytic layer Co_3_O_4_ had no obvious influence on the strength and tensile deformation, mainly because the supporting layer was thin. In addition, the average Young’s modulus of Co_3_O_4_/CP is 22.9 MPa, which is not significant compared to the 19.4 MPa of commercial CP (Appendix A). This indicates that the loading of Co_3_O_4_ and the calcination process have little effect on the stiffness of carbon fiber paper, and Co_3_O_4_/CP can retain the original flexibility.

The improvement of the mechanical properties of the Co_3_O_4_/CP material were attributed to the increase in the interwoven structure of the chopped carbon fibers and the point bonding effect of the adhesive. Co_3_O_4_/CP exhibits a higher mesoporous content structure, with a mesoporous diameter between 2 and 5 nm (Figure 5d). The porous structure of Co_3_O_4_/CP can buffer the volume change of the electrode during charge and discharge, which is a benefit for improving the electrochemical performance of the electrode [8].

Based on the excellent flexibility and high-efficiency catalytic performance of the Co_3_O_4_/CP electrode, a flexible rechargeable zinc-air battery was prepared. The open circuit potential (OCP) of the flexible ZAB with a Co_3_O_4_/CP electrode measured by the electrochemical workstation can reach 1.34 V (as shown in Figure 6a), which is consistent with the value reported in the literature for the zinc-air battery. Figure 6b shows the charging and discharging electric polarization curve of the flexible ZAB. Compared with ZAB using commercial CP as the air cathode, ZAB using Co_3_O_4_/CP air cathode exhibits a lower overpotential during charging and discharging. The gap becomes more obvious with the increase in current density.

Figure 6c show the discharge performance of different flexible zinc-air batteries under varying current density conditions. With the increase of the discharge current density, the discharge voltage of ZAB decreased significantly. Compared with commercial CP-ZAB, Co_3_O_4_/CP-ZAB has a higher discharge voltage at the same current density. This difference was more pronounced at high current densities. The reasons for the excellent electrocatalytic performance of Co_3_O_4_/CP-ZAB are as follows: the tight nanostructure load between the Co_3_O_4_ layer and the surface of the carbon fiber and the abundant mesoporous structure provide a channel for the rapid transmission of electrons during the catalytic reaction; the close combination of carbon fiber active material and gel electrolyte; the tight three-dimensional interweaving structure of carbon fiber improves its electrical conductivity. In order to further study the stability of Co_3_O_4_/CP electrodes during cycling, ZAB was tested with constant current charge and discharge. As shown in Figure 6d, compared to commercial CP, Co_3_O_4_/CP ZAB exhibits a lower charge and discharge overpotential, which means that the battery is more rechargeable. In addition, at a current density of 2 mA·cm^−2^, commercial CP ZAB cycle time is less than 36 h, and the cycle number is about 100 times (Appendix A). The Co_3_O_4_/CP air cathode ZAB can continuously cycle and stably charge and discharge for more than 60 h, and the number of cycles can reach more than 180 times. Compared with other carbon fiber substrate cathode research in recent years, the Co_3_O_4_/CP flexible ZAB still shows good stability (as shown in Appendix A).

The flexible zinc-air battery was subjected to long-term charge–discharge cycle tests under bending angles of 0°, 60°, and 180°. As shown in Figure 7a, the results proved that ZAB assembled with Co_3_O_4_/CP as the base still had a relatively consistent long-term work in a highly bending working environment. Life, and the voltage range, can also be stabilized at 1.0~2.0 V. This proves that the laminated zinc-air battery has a relatively excellent flexible working performance. It can be seen from Figure 7b that, at any given bending angle, the manufactured ZAB has excellent stability without significant changes in potential. In addition, at a current of 2 mA·cm^−2^, the highly curved flexible ZAB can still be discharged stably for 33.7 h, with a discharge capacity of 67.4 mAh·cm^−2^. This demonstrates the excellent energy conversion performance of the Co_3_O_4_/CP flexible ZAB.

According to the Co_3_O_4_/CP-ZAB preparation and assembly process described above, three ZABs encapsulated by breathable gauze and aluminum plastic film are connected in series. As shown in Figure 8a, the three ZAB groups exhibited a relatively high 4.12V open circuit voltage. Wearing it as a flexible device on the hand, its output voltage basically remains unchanged (Figure 8b). In addition, these ZABs can be used to charge light-emitting diode (LED) screens through a serial bus transmission interface (Figure 8c). This proves that it has good wearable energy storage device performance.

## 4. Conclusions

In summary, an economical and efficient method is developed for preparing flexible cathodes. In this work, a dense mesoporous Co_3_O_4_ layer was first hydrothermally grown in situ on the surface of a single chopped carbon fiber (CF), and then carbon fiber paper (Co_3_O_4_/CP) was prepared by a wet papermaking process as a flexible zinc-air battery. Thanks to the high catalytic performance of the Co_3_O_4_ layer and the high conductivity of the active material, the quality activity of the Co_3_O_4_/CP electrode for oxygen reduction and precipitation reactions is far higher than that of commercial carbon paper of the same quality. Through the wet papermaking process, Co_3_O_4_/CP has a high mechanical stability and excellent electrical conductivity. In addition, the assembled zinc-air battery exhibits excellent electrochemical performance, with a continuous cycle of more than 180 times at a current density of 2mA·cm^−2^. This traditional papermaking method provided a new pathway for designing flexible air-cathodes. Combining the uniform loading of the Co_3_O_4_ catalytic layer with the carbon fiber network integration process improves the uniformity of internal loading and improves process continuity. Consequently, this paper-based air cathode would hold considerable potential for rechargeable flexible ZAB applications.

## Data Availability

The data presented in this study are available on reasonable request from the corresponding author.

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
