# Peer review of "Co3O4 Nanoneedle Array Grown on Carbon Fiber Paper for Air Cathodes towards Flexible and Rechargeable Zn–Air Batteries"

_nanomaterials, 2021, doi:10.3390/nano11123321_

Round 1
Reviewer 1 Report
The work presented to me for review "Co3O4 Nanoneedle Array Grown on Carbon Fiber Paper for Air Cathodes towards Flexible and Rechargeable Zn–Air Batteries" by Ziyuan Li, Wenjia Han, Peng Jia, Xia Li, Yifei Jiang, Qijun Ding, is not a work of great originality. The idea of such research was already known. The authors added their contribution to the issue studied by other authors. In order to obtain flexible cathodes as a substrate, instead of carbon cloth Co3O4/CC (X. Chen et al. Advanced Energy Materials. 7(18) (2017) 1700779) they used carbon fiber paper Co3O4/CP.
The authors did not avoid carelessness in the editing of the work.
line 57 should be: oxygen reduction reaction (ORR),
figure 1(b) too small markings in the figure,
line 93 remove one (Figure 4b),
figure 5: wrong figure 5(a) , correct ; line 210, line 218, line 252
Reviewer 2 Report
In this manuscript, the authors report their implementation of a zinc-air battery where the cathode is composed of carbon fibers paper impregnated with cobalt oxide. The new material was characterized by electron microscopy, EDS, XRD, XPS, Raman, thermogravimetry, mechanical stress analysis, and electrochemical experiments.
The authors also showed that a zinc-air battery made from their proposed cathode material can work also in the bent configuration as a wearable device.
The work is interesting and some of the results are promising, but a lot of details are not given. The writing should also be double-checked. Although some of the results are good the manuscript requires a large amount of work to meet the scientific standard.
-
Are Zinc-air batteries secondary batteries?
-
What is the CO2 etching?
-
Please provide more information for the electrochemical experiments: reference electrode, potential scale, scan rate…
-
Only the oxygen reduction reaction was shown, what about the complementary reaction to charge the battery?
-
There are electrochemical impedance spectra in the supporting information, as may be indicated in the main text at line 201.
-
In the supporting information, there are only figures and tables with no description. This is not a good practice.
-
Please provide an analysis of the electrocatalytic effect from the rotating disk experiments.
-
About figure 6:
-
part b shows only positive currents instead of positive and negative ones.
-
What are the current density values in part c?
-
What is the current density of part d? How was this experiment performed? Clearly, there was no voltage limit enabled. The discussion made on this point is also misleading. Clearly, the battery with the CP has larger voltages which are good for a battery. The authors should calculate the charge efficiency, voltage efficiency, and energy efficiency to make a fair comparison.
-
Part e displayed in this way is not useful. The authors should calculate a charge capacity or another cycle-related figure of merit and plot this against the time or number of cycles.
-
-
Please see the attached file for some language mistakes/error I found.

Reviewer 3 Report
The manuscript reports an in situ hydrothermal synthesis of a dense mesoporous Co3O4 layer onto a single chopped carbon fiber (CF) surface, followed by the preparation of a carbon fiber paper (Co3O4/CP) by a wet papermaking process. In this work, the Co3O4/CP is employed as a flexible cathode of Zinc-air battery. However, Co3O4 on carbon paper materials have been studied as flexible catalyst for air cathode already, e.g. Ref. [30]. And there are many other previous literatures that report Co3O4 bifunctional catalyst, e.g., https://pubs.rsc.org/en/content/articlelanding/2017/nr/c7nr02385e Therefore, the novelty of this manuscript lies on the improved mechanical strength and electrical conductivity. However, I find that those novel points have not been fully addressed in this present form.
- Measurement and comparison of actual electronic conductivity of the samples are necessary.
- More detailed analysis on the mechanical properties are required. The picture of Figure 5a seems that it is still stiff in many positions. Also, Young’s modulus of the newly developed CP is higher (i.e., more stiff) than the commercial CP. Please discuss in more detail, with detailed values of modulus and maximum strength. The values should be reported in the form of average +- standard deviation for statistically sound results.
- For the following statement, “Unfortunately, the low discharge capacity of fibrous batteries limits its application in the field of flexible energy storage”, in lines [76 - 78], please cite some references to support the argument.
- In the section 2.1 of experimental section, the authors have mentioned “…were modified via the CO2 etching at 500°C for one hour, aiming at removing the impurities, improving the hydrophilicity and increasing the nucleation site of Co(OH)2…”. Apart from the SEM images in S1, please provide other evidence for showing improvement in hydrophilicity (like contact angle) and nucleation site of Co(OH)2. Also, provide some references.
- In section 2.2 of the experimental section, please define the acronym ‘PVA’.
- Please cite some references to support the statement in line 188, “…These peaks are characteristic of Co3O4…”.
- In line 205, the authors have stated: “…In Figure S4b, the peak current density of the oxi-205 dation-reduction reaction of the Co3O4/CP electrode…”. To have a better understanding of the ORR, please provide information on the reference electrode used to conduct the CV measurements.
- The authors have stated “…its electrochemical stability and working efficiency are 275 also very close to traditional liquid batteries…” in line 275. To support this claim, please cite some references and prepare a comparison table.
- There are numerous typos and grammatical errors that I have amended in the attached PDF file. Please rectify the errors.

Round 2
Reviewer 2 Report
Thank you for the clarifications and the modifications.